# Evaluation of HIV Transmission Clusters among Natives and Foreigners Living in Italy

**DOI:** 10.3390/v12080791

**Published:** 2020-07-23

**Authors:** Lavinia Fabeni, Maria Mercedes Santoro, Patrizia Lorenzini, Stefano Rusconi, Nicola Gianotti, Andrea Costantini, Loredana Sarmati, Andrea Antinori, Francesca Ceccherini-Silberstein, Antonella d’Arminio Monforte, Annalisa Saracino, Enrico Girardi

**Affiliations:** 1Laboratory of Virology, National Institute for Infectious Diseases “Lazzaro Spallanzani” IRCCS, 00149 Rome, Italy; lavinia.fabeni@inmi.it; 2Department of Experimental Medicine, University of Rome “Tor Vergata”, 00133 Rome, Italy; ceccherini@med.uniroma2.it; 3Clinical Department, National Institute for Infectious Diseases “Lazzaro Spallanzani” IRCCS, 00149 Rome, Italy; patrizia.lorenzini@inmi.it (P.L.); andrea.antinori@inmi.it (A.A.); 4III Division of Infectious Diseases, ASST Fatebenefratelli Sacco, Luigi Sacco University Hospital, 20157 Milan, Italy; stefano.rusconi@unimi.it; 5Infectious Diseases, San Raffaele Scientific Institute, 20127 Milan, Italy; gianotti.nicola@hsr.it; 6Department of Clinical and Molecular Science, Marche Polytechnic University, 60121 Ancona, Italy; a.costantini@univpm.it; 7Clinical Infectious Diseases, Policlinic Foundation Tor Vergata, 00133 Rome, Italy; sarmati@med.uniroma2.it; 8Department of Health Sciences, University of Milan, 00161 Milan, Italy; antonella.darminio@unimi.it; 9Department of Biochemical Sciences and Human Oncology, University of Bari, 70121 Bari, Italy; annalisa.saracino@uniba.it; 10Clinical Epidemiology Unit, National Institute for Infectious Diseases “Lazzaro Spallanzani” IRCCS, 00149 Rome, Italy; enrico.girardi@inmi.it

**Keywords:** human immunodeficiency virus (HIV), molecular epidemiology, phylogenetic analysis, migrants, cluster detection, transmission networks and clusters, subtypes, drug resistance testing, risk factors, bioinformatics

## Abstract

We aimed at evaluating the characteristics of HIV-1 molecular transmission clusters (MTCs) among natives and migrants living in Italy, diagnosed between 1998 and 2018. Phylogenetic analyses were performed on HIV-1 polymerase (*pol*) sequences to characterise subtypes and identify MTCs, divided into small (SMTCs, 2–3 sequences), medium (MMTCs, 4–9 sequences) and large (LMTCs, ≥10 sequences). Among 3499 drug-naïve individuals enrolled in the Italian Cohort Naive Antiretroviral (ICONA) cohort (2804 natives; 695 migrants), 726 (20.8%; 644 natives, 82 migrants) were involved in 228 MTCs (6 LMTCs, 36 MMTCs, 186 SMTCs). Migrants contributed 14.4% to SMTCs, 7.6% to MMTCs and 7.1% to LMTCs, respectively. HIV-1 non-B subtypes were found in 51 MTCs; noteworthy was that non-B infections involved in MTCs were more commonly found in natives (*n* = 47) than in migrants (*n* = 4). Factors such as Italian origin, being men who have sex with men (MSM), younger age, more recent diagnosis and a higher CD4 count were significantly associated with MTCs. Our findings show that HIV-1 clustering transmission among newly diagnosed individuals living in Italy is prevalently driven by natives, mainly MSM, with a more recent diagnosis and frequently infected with HIV-1 non-B subtypes. These results can contribute to monitoring of the HIV epidemic and guiding the public health response to prevent new HIV infections.

## 1. Introduction

Although HIV-1 is preventable through effective public health measures, substantial HIV transmission continues to occur worldwide. In 2018, 1.7 million HIV-1 newly infected individuals were reported [1]. In particular, 141,552 newly diagnosed HIV infections were reported in 50 of the 53 countries in the European Region of the World Health Organization (WHO) [2], while 2847 new cases were reported in Italy [3].

Among the HIV-1-diagnosed individuals in the EU/EEA in 2018, 42% were migrants, defined as originating from outside of the country in which they were diagnosed [2]. In Italy, migrants accounted for nearly 30% of all newly diagnosed HIV infections in recent years [2,3]. As a consequence of the high percentage of migrants in several European countries including Italy, the prevalence of numerous HIV-1 non-B subtypes and circulating recombinant forms (CRFs) has rapidly increased in several previously B-restricted areas [4,5,6,7,8,9]. However, the increasing prevalence of non-B subtypes is only partially due to migrants, since a substantial amount of such increase is attributable to the European-born population, including Italians [4,5,7,8,9,10].

A considerable proportion of the new HIV diagnoses is involved in well-defined clusters prevalently characterized by high-risk sexual behaviours [6,8,11,12,13,14]. Accurate identification of these transmission clusters and an understanding of HIV-transmission dynamics are very important to efficiently target public health interventions. In this context, phylogenetic analysis represents one of the most important strategies to better describe and monitor the local HIV epidemics. When combined with epidemiological and clinical data, the results of such analysis can be of public health relevance, for example by identifying how virus lineages are restricted to, or mix among, different demographic and behavioural subgroups [15,16].

The aim of this study was to evaluate molecular transmission clusters (MTCs) among native and foreign individuals living in Italy diagnosed between 1998 and 2018. Characteristics and drivers of the HIV-1 epidemic in Italy were also evaluated by combining MTCs analysis with demographic, epidemiological and laboratory data.

## 2. Materials and Methods

### 2.1. Cohort Description

The Italian Cohort Naive Antiretroviral (ICONA) Foundation Study is a multi-centre prospective observational cohort of HIV-1-infected patients, which was set up in 1997 [17]. Eligible patients are those starting combined antiretroviral therapy when they are naive to antiretrovirals, regardless of the reason for which they had never been previously treated and of the stage of their disease. All patients sign a consent form to participate in ICONA, in accordance with the ethical standards of the committee on human experimentation and the Helsinki Declaration (1983 revision). Demographic, clinical and laboratory data and information on therapy are collected for all participants and recorded using electronic data collection; sensitive data are collected only in an anonymous format. Subjects enrolled in the ICONA cohort are broadly representative of the Italian HIV-diagnosed population [18].

### 2.2. Study Population

For this specific study, 3499 newly diagnosed patients enrolled in the ICONA cohort during the period between 1998 and 2018, for whom an HIV-1 *pol* sequence was available at the time of the diagnosis, were considered for the analysis. Individuals born in a country other than Italy were classified as migrants. *Pol* sequences (containing the full-length protease (PR) and the first 240/335 reverse transcriptase (RT) codons) were produced from plasma samples through the classical Sanger sequencing performed according to the protocols used at the laboratories at each Clinical Center contributing to the cohort.

### 2.3. Phylogenetic Analyses

Molecular phylogeny was used to determine both HIV-1 subtype for each sequence, as previously described [19], and for the cluster analysis. MTCs were first deduced using the HIV-TRACE (Transmission Cluster Engine) web tool [20,21]. Pairwise genetic distances were obtained using HIV-TRACE under the Tamura-Nei 93 (TN93) nucleotide substitution model. This is the most general nucleotide substitution model for which distances can be estimated directly from counts of nucleotide pairs in aligned sequences [21]. Putative transmission links were inferred when two sequences had TN93 genetic distance ≤0.010 substitutions/site. This stringent genetic distance threshold has been chosen according to the suggestions by Wertheim and colleagues [22], based on the consideration that within a single person, HIV *pol* sequences tend not to diverge more than 0.01 substitutions/site from the baseline sequence in the first 10 years of infection [23], with the total sequence divergence tending to be less than 0.02 substitutions/site [24]. Therefore, one would expect a meaningful epidemiologically genetic distance threshold for identifying transmission partners to fall between 0.01 and 0.02 substitutions/site.

To avoid the influence of convergence evolution at antiretroviral drug resistance mutations, sequences were stripped at positions related to drug resistance.

The robustness of the MTCs was further tested using the maximum likelihood (ML) method and a Bayesian analysis. MTCs observed by HIV-TRACE analysis were not considered if not retrieved by ML and/or Bayesian phylogenetic analysis. The ML tree was inferred as previously described, using the MEGA version 6.06 software [25]. MTCs were recognised by bootstrap values >90% and a <0.015 average genetic distance. Finally, the Bayesian phylogenetic tree was reconstructed with MrBayes version 3.2.2, using a GTR + I + Г_5_. The Monte Carlo Markov Chain search was run for 5 × 10^6^ generations, with the trees sampled every 100th generation (with a burn-in of 50%) [26]. Statistical support was obtained by calculating the posterior probability of each monophyletic clade, and a posterior consensus tree was generated after 50% burn-in. Clades were considered epidemiological clusters only if a posterior probability of ≥0.90 was inferred.

MTCs were then divided into small clusters (SMTCs, 2–3 sequences), medium clusters (MMTCs, 4–9 sequences) and large clusters (LMTCs, ≥10 sequences).

### 2.4. Evaluation of HIV-1 Transmitted Drug Resistance

Transmitted drug resistance (TDR) was evaluated by considering the WHO 2009 list [27] with the additional RT mutations K65E/N, E138A/G/K/Q/R, V179L, G190Q, T215N, H221Y, F227C and M230I, reported in the International AIDS Society list [28] and/or the HIVdb V.8.9-1 (https://hivdb.stanford.edu/, last updated 25/10/2019). HIV-1 strains were defined as resistant if carrying at least one TDR mutation.

### 2.5. Statistical Analysis

Comparisons between patients belonging or not to clusters were evaluated using the Mann–Whitney test for quantitative variables and Chi-square or Fisher’s exact test for categorical variables. Factors potentially associated with MTCs were evaluated using multivariable logistic regression analysis. In particular, the following factors were included in the model: gender, mode of HIV transmission, nation of birth, level of education, type of employment, plasma HIV RNA and CD4 cell count at sequencing and calendar year of HIV diagnosis. All analyses were performed using STATA version 15.1 (StataCorp College Station, Texas, USA). *p*-values <0.05 were considered as statistically significant.

### 2.6. Sequences Deposited to GenBank

All 726 sequences involved in the 228 molecular transmission clusters have been deposited into GenBank (accession numbers: MT787750-MT788351; MT788352-MT788382; MT788383-MT788475). All other sequences are available on request.

## 3. Results

### 3.1. Patients’ Characteristics

A total of 3499 HIV-1 newly diagnosed participants in the ICONA cohort from 1997 through 2018 were included (Table 1). The majority of these individuals (2872; 82.1%) were male, Italian (2804; 80.1%) and MSM (1789; 51.1%). Migrants accounted for 19.9%, had lived in Italy for a median (interquartile range, IQR) of 5 (1–9) years and were represented mainly by subjects from Central and South America (241, 6.9%) and Africa (219, 6.3%), followed by Europe (187, 5.3%), Asia (38, 1.1%) and other areas (10, 0.3%) (Table 1 and Appendix A). Phylogenetic analysis revealed that most individuals (*n* = 2556; 73.1%) were infected with B subtype, followed by CRF02_AG (187; 5.3%), F1 (179; 5.1%), C (148; 4.2%), A1 (104; 3.0%), CRF60_BC (64; 51.8%) and other subtypes/CRFs (261; 7.5%).

The overall prevalence of TDR was 14.3% (*n* = 501), mainly imputable to resistance associated with non-nucleoside RT inhibitors (NNRTI) (9.6%).

### 3.2. Cluster Identification

Overall, 228 MTCs including 2–52 individuals were identified by both HIV-TRACE and phylogenetic analyses (Figure 1, Appendix A). No differences between the two methods were identified. MTCs involved 726 individuals (20.7%; 644 natives and 82 migrants). Of the 228 MTCs identified, 186 were SMTCs (2–3 sequences) involving 402 individuals (11.5%), 36 MMTCs (4–9 sequences) involving 184 individuals (5.3%) and 6 LMTCs (≥10 sequences) involving 140 individuals (4.0%) (Table 2). The presence of both natives and migrants was found in 66.7% of LMTCs, 33.3% of MMTCs and 23.1% of SMTCs. Migrants were involved mainly in SMTCs (*n* = 58; 70.7%). Individuals infected with HIV-1 B subtype in MTCs were present with a higher prevalence than those infected with non-B subtype (71.4% vs. 28.6%). Individuals infected with non-B subtypes were part of 51 MTCs (22.4%) (SMTCs: 37; MMTCs: 11; LMTCs: 3; Table 2); only four of these MTCs were entirely composed of migrants (7.8%).

By analyzing the date of diagnosis of individuals involved in MTCs, it was found that before 2010 the proportion of individuals involved in the clusters was 34.6% (*n* = 251: SMTCs, *n* = 156, 38.8%; MMTCs/LMTCs = 95, 29.3%). Only a few individuals were involved in MTCs before 2007 (*n* = 38, 5.2%: SMTCS, *n* = 30, 7.5%; MMTCs/LMTCs = 8, 2.5%). Starting from 2011, the proportion of individuals involved in MTCs was 65.4% (*n* = 475: SMTCs, *n* = 246, 61.2%; MMTCs/LMTCs = 229, 70.7%). Migrants, on the other hand, were only involved in MTCs from 2007. In particular, the proportion of migrants in MTCs between 2007 and 2010 was overall 26.8% (*n* = 22: SMTCs, *n* = 18, 31.0%; MMTCs/LMTCs = 4, 16.7%), whereas after 2011 the proportion of migrants in MTCs was 73.2% (*n* = 60: SMTCs, *n* = 40, 69.0%; MMTCs/LMTCs = 20, 83.3%).

Regarding TDR, 81 (11.2%) individuals involved in MTCs carried a resistant virus. This proportion was lower compared to that found in individuals out of MTCs (15.1%, *p* = 0.006). Individuals with a resistant virus were involved in 36 SMTCs (29 B, 2 C, 2 CRF01_AE, 1 CRF02_AG, 1 CRF24_BG, 1 F1), five MMTCs (three B, one CRF02_AG, and one CRF12_BF) and one LMTC (CRF60_BC).

### 3.3. Epidemiological Characteristics and Factors Associated with MTCs

With regard to the characteristics of individuals involved in MTCs in comparison with those out of MTCs, male gender (94.3% vs. 78.9%, *p* < 0.001), being MSM (75.8% vs. 45.0%, *p* < 0.001) and natives (88.7% vs. 77.8%, *p* < 0.001) were significantly associated with being in MTCs (Table 1). Individuals in MTCs were also younger (median (IQR) age: 32 (27–40) vs. 38 (31–46) years, *p* < 0.001), more recently diagnosed (median (IQR) year of diagnosis: 2012 (2009–2014) vs. 2011 (2007–2014), *p* < 0.001), and had higher CD4 cell count on diagnosis (median (IQR) cells/mm^3^: 459 (322–624) vs. 353 (177–523), *p* < 0.001) in comparison with individuals out of MTCs. Moreover, a lower prevalence of TDR was found among individuals in MTCs compared to those out of MTCs (11.2% vs. 15.1%, *p* < 0.001). Multivariable logistic regression confirmed that Italian origin, being MSM, of younger age, a more recent diagnosis, and higher CD4 cell count were factors significantly associated with MTCs (Table 1).

### 3.4. Characteristics of the Medium/Large Clusters

Information of MMTCs (*n* = 36) and LMTCs (*n* = 6) is reported in Table 2 and Table 3. The majority of MTCs were composed only of males (MMTCs: 32/36, 88.9%; LMTCs: 5/6, 83.3%). The most common risk factor was same-gender sex (49.8% of the MSM in the overall MTCs), followed by heterosexual contacts (24.8% of the heterosexuals in the overall MTCs). As regards to migrants, they contributed 7.6% to MMTCs and 7.1% to LMTCs, respectively. The 24 migrants involved in these medium/large MTCs were mainly from Central/South America or other European countries (Table 3).

Some of the individuals involved in these MTCs carried the following pathways of mutations associated with drug resistance: I85V in PR (*n* = 1), and A62V (*n* = 3), E138A (*n* = 6) or K101E (*n* = 5) in RT (Table 3). The median genetic distance was ≤0.016 substitution/site in each MTC, highlighting the tight correlation among individuals involved in the MTC.

### 3.5. Characteristics of the Small Clusters

Focusing attention on the 186 SMTCs, 156 (83.9%) were composed of two persons, and 30 (16.1%) of three persons (Table 2). The contribution of females in these SMTCs was minimal (34/402 (8.5%): 31 in pairs and 3 in MTCs composed of three individuals). Overall, these SMTCs were composed mainly of MSM (54.3%). Interestingly, 32.8% included individuals with mixed transmission categories, mainly among MSM and heterosexual contacts (19.9%) predominantly found in TCs (86.5%) pairs. The great majority of these TCs were exclusively composed of Italians (73.7%), while about 23.1% of them were composed of mixed nationality. Only six (3.2%) of the SMTCs were totally composed of migrant couples (Table 2) originating from Africa (*n* = 3), Asia (*n* = 1), America (*n* = 1) and Eastern Europe (*n* = 1) (data not shown); four of them (67%) were composed of heterosexual individuals that came from the same country in Africa (*n* = 3) and Asia (*n* = 1) (Figure 1), while the other two were composed of MSM.

Regarding TDR, 66 individuals (involved in 36 SMTCs, 19.3%) carried drug resistance mutations.

In particular, seven SMTCs (12 individuals) carried only PI resistance mutations (K23I, M46I/L, F53Y, I85V, L90M); twenty-four SMTCs (43 individuals) carried NNRTI resistance mutations (L100I, K101E, K103N/S, V106I, E138A/G/K, G190A), alone or in combination with other RTI resistance mutations. Specifically, one SMTC, composed of two individuals, carried the combination K103N+E138A, and one SMTC had the combination L100I + K103N + L210W + T215D.

Five SMTCs (11 individuals) carried NRTI resistance mutations (M41L, D67N, L210W, T215S/L/D, K219Q), alone or in combination with other NRTI mutations. In particular, the following combinations were found in all the individuals involved in the MTCs: M41L+T215L (three individuals), D67N+K219Q (two individuals), while one pair was composed of one individual with the M41L and one with the M41L+T215D combination.

## 4. Discussion

This study provides molecular evidence of HIV-1 epidemiological clusters circulating in Italy among 3499 native and foreign individuals diagnosed between 1998 and 2018 enrolled in the ICONA cohort. Around 21% of the overall individuals included in the analysis were involved in clusters. Clustering analysis showed the presence of a high number of MTCs; in particular, 228 MTCs including 6 LMTCs, 36 MMTCs and 186 SMTCs were identified.

In general, clustered transmission was prevalently driven by native individuals (644, 88.7%); in fact, 222 MTCs (97.4%) were composed mainly of Italian individuals, and more in particular, 157 MTCs (70.7%) were composed exclusively of Italian persons and 65 MTCs (29.3%) showed an intermixing between migrants and Italians. Only six SMTCs of the overall MTCs were composed exclusively of a couple of migrant individuals—67% were composed by heterosexual individuals and came from the same country in Africa and Asia. This highlights how some migrants probably acquired HIV in their home country and how in certain geographical areas the transmission route is mainly represented by heterosexual contacts [29]. A minimal contribution of migrants in MTCs was also found in Belgium, despite the high prevalence of foreign individuals living in this country [10]. In particular, in the Belgian cohort, the percentage of foreign people was much higher than in our cohort (53.0% vs. 19.9%). Furthermore, foreign individuals infected with HIV (for example those from Europe) might look slightly less actively involved in cluster formation in Italy than in Belgium, although both cohorts showed similar tendencies. Despite these differences between the two cohorts, overall findings support the fact that clustering is mainly driven by natives.

Individuals involved in MTCs were younger, mainly male and MSM, with a recent diagnosis and higher CD4 count compared to those out of MTCs. These findings are in agreement with several studies on transmission clusters from the USA, Asia and Europe [8,10,14,19,30,31,32,33,34,35,36] and led us to hypothesise that MSM are more aware than others of having both high-risk behaviour and a high self-perception of risk for acquiring HIV, and for this reason, they test more frequently than others and are thus diagnosed earlier.

Even though MSM was the most prevalent risk factor in MTCs, different risk behaviours were observed among individuals belonging to the same MTCs. In particular, mixtures of MSM and heterosexual contacts were observed in most MTCs. In fact, at least one heterosexual contact was observed in 19.9%, 27.8%, and 83.0% of each SMTCs, MMTCs, and LMTCs, respectively. Noteworthy was that the majority of MTCs found the number of MSM clearly outweighed the number of heterosexuals, and in many, only men were represented. This observation is in line with that reported in previous studies and suggests that misreporting of transmission risk may be an important reason for the presence of mixed clusters [10,37,38]. The most important reason for misreporting could be fear or embarrassment to disclose sex with same-gender sex. Another explanation for the presence of mixed transmission risk clusters may be due to the fact that homosexual orientation is frequently masked by bisexual behaviours in older individuals, and men who have sex with men and women are a potential bridge population for transmitting HIV to heterosexual women [6,10,39,40]. Along the same line, intravenous drug users may serve as a go-between risk category.

Of note, about a quarter of MTCs were characterized by HIV-1 non-B subtypes; in particular, 50% of LMTCs and 30% of MMTCs were driven by these viral strains. By considering the fact that only a small portion of migrants was present in these non-B MTCs, in the most part of cases interspersed with natives, this finding confirms the fact that HIV-1 non-B variants have entered and are to date circulating in newly diagnosed individuals originating from Italy. This is in line with the important increase of non-B pure subtypes and CRFs reported in several European countries during the last decade. This increase in non-B subtypes occurred in conjunction with associated epidemiological changes, such as (1) increase in immigration and tourism from areas with non-B subtypes, (2) travel-associated infections and (3) increased internal circulation in Italy of imported subtypes [5,7,8,9,10,12,14,41,42,43,44,45,46,47].

By specifically considering the date of diagnosis, most individuals in MTCs were diagnosed after 2010, thus suggesting that the expansion of epidemics in Italy seems mainly associated with recent MTCs, rather than the growth of older, established ones. This has also been confirmed by multivariable logistic regression analysis, showing that a more recent diagnosis is a factor positively associated with MTCs. The involvement of migrants in MTCs has a similar temporal trend of the overall population. From the public health point of view, all this information is important for designing prevention and public health intervention strategies.

Regarding TDR, 14.3% of the overall individuals harboured a resistant virus, mainly imputable to NNRTI resistance mutations. This TDR prevalence is slightly higher than in other studies [7,39,46,48] because in the analysis, we considered the E138A mutation, associated with resistance to etravirine and rilpivirine [49,50,51,52]. In fact, by removing this mutation, the overall prevalence of TDR in our study population was found to be 9.8%, similar to that already reported in literature [7,10,42,46,48,53], remaining mainly attributed to NNRTI resistance (4.8%). As E138A is a polymorphic mutation that ranges in prevalence from about 1% to 5% depending on subtype [54], its high prevalence in our population might not necessarily be related to TDR.

By evaluating the prevalence of TDR in and out of MTCs, similarly to that observed in previous studies [8,46], a resistant virus was less frequent among individuals involved in MTCs compared with those out (11.2% vs. 15.1%), and this was also confirmed by excluding the E138A mutation (data not shown). On the other hand, different results have been found by Verhofstede et al. [10] describing a slightly higher TDR prevalence within the clusters. These discrepancies might be explained by different characteristics of the study populations analyzed. Resistance to more than one class was found only in one SMTCs, where mutations for both NRTIs and NNRTIs were present. By assessing TDR prevalence in MTCs according to drug class, we mainly found the mutations associated with resistance to NNRTIs K101E and E138A, which are at low fitness cost [55,56].

As with any other observational study, our data may have some limitations. Firstly, our study population might not be representative of the overall population living in Italy. In particular, it has been estimated that to date, migrant individuals with new HIV-1 diagnoses accounted for nearly 30% of all the newly diagnosed HIV infections in Italy in recent years [3]. Despite the fact that in the overall ICONA cohort this percentage is confirmed [17], in our specific study population, the overall percentage of migrants was lower (around 20%). This is due to the fact that the sequences at diagnosis were not available for all the individuals enrolled in the cohort. Thus the contribution of migrants in MTCs could be underestimated; in fact, unsampled individuals might limit the correct idea of the migrant circulation in our country. However, our results are in line with other previous Italian studies in different study populations that showed that the contribution of migrants in the cluster is minimal [6,8,46].

Moreover, we should take into account that phylogenetic analyses are limited by their inability to infer direction of transmission: the absence of the contribution of sequences from drug-treated individuals and unsampled individuals might limit correct transmission dynamics. Finally, another significant limitation to this study is the lack of information on the country of infection, because these are important data to establish the geographical origin of the MTCs. This could explain, for example, the presence of couples in our study population composed only of heterosexual individuals of the same gender, even though we cannot exclude a misclassification of the risk factor for these cases. Nonetheless, the detection of MTCs and the associated demographic and viroimmunological parameters can help us to focus proven HIV prevention tools where they are needed most.

In conclusion, this study shows that HIV-1 newly diagnosed subjects are involved in several MTCs in the past two decades in Italy. Clustered transmission, especially for large clusters, is prevalently driven by natives, mainly MSM, with a recent diagnosis and frequently infected with HIV-1 non-B subtype. Participation of migrants in clustered transmission is rare.

These results reinforce the fact that phylodynamics represent one of the most important tools to better describe and monitor local HIV epidemics, by correlating the genetic relationship of the viruses with information on demographics, transmission mode and new infections. 

Overall, our findings can contribute to monitoring of the HIV epidemic and guiding the public health response in Italy to prevent new infections.

## Figures and Tables

**Figure 1 viruses-12-00791-f001:**
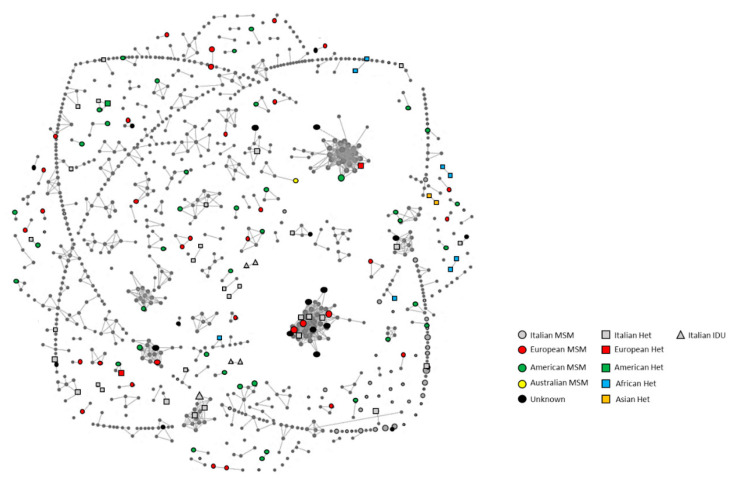
Clusters’ population by HIV-TRACE. The coloured geometric shapes identify the nationality and risk factors of the 3499 individuals involved in the study. Shapes without connections represent individuals not involved in molecular transmission clusters. Shapes connected with lines represent individuals involved in molecular transmission clusters.

**Table 1 viruses-12-00791-t001:** Patient’s characteristics and factors associated with HIV-1 molecular transmission clusters.

Variables	Overall	Out of Cluster	In Cluster	*p*-Value ^a^	Adjusted Model ^b^		
	*n* = 3499	2773 (79.3%)	726 (20.7%)		OR	(95% CI)	*p*-Value
**Male gender, *n* (%)**	2872 (82.1%)	2187 (78.9%)	685 (94.3%)	<0.001	-	-	-
**Age, years, median (IQR)**	37 (30–45)	38 (31–46)	32 (27–40)	<0.001	0.65	0.59–0.72	<0.001
**Mode of HIV transmission, *n* (%)**							
F heterosexual	553 (15.8%)	513 (18.5%)	40 (5.5%)	<0.001	1.00	-	-
F IVDU	32 (0.9%)	31 (1.1%)	1 (0.1%)		0.49	0.06–3.80	0.497
M heterosexual	713 (20.4%)	628 (22.7%)	85 (11.7%)		1.82	1.19–2.78	0.006
M IVDU	161 (4.6%)	145 (5.2%)	16 (2.2%)		1.52	0.80–2.90	0.204
MSM	1789 (51.1%)	1247 (45.0%)	550 (75.8%)		3.46	2.39–5.03	<0.001
Other/unknown	251 (7.2%)	209 (7.5%)	34 (4.7%)		2.73	1.66–4.48	<0.001
**Nation of birth, *n* (%)**							
Italy	2804 (80.1%)	2160 (77.8%)	644 (88.7%)	<0.001	1.00	-	-
Africa	219 (6.3%)	212 (7.7%)	7 (0.9%)		0.18	0.08–0.39	<0.001
Central and South America	241 (6.9%)	201 (7.3%)	40 (5.5%)		0.49	0.33–0.71	0.001
Europe	187 (5.3%)	159 (5.7%)	28 (3.9%)		0.29	0.08–0.97	0.045
Asia	38 (1.1%)	35 (1.3%)	3 (0.4%)		0.62	0.40–0.97	0.035
Other	10 (0.3%)	6 (0.2%)	4 (0.6%)		2.61	0.62–10.97	0.189
**Education, *n* (%)**							
Primary school	169 (4.8%)	158 (5.7%)	11 (1.5%)	<0.001	0.87	0.45–1.71	0.691
Secondary school	585 (16.7%)	505 (18.2%)	80 (11.0%)		0.91	0.68–1.21	0.518
College/University	1762 (50.4%)	1329 (47.9%)	433 (59.6%)		1.00	-	
Unknown	983 (28.1%)	781 (28.2%)	202 (27.8%)		1.03	0.83–1.28	0.773
**Employment, *n* (%)**							
Employed	1476 (42.2%)	1148 (41.4%)	328 (45.2%)	<0.001	1.00	-	-
Unemployed	461 (13.2%)	389 (14.0%)	72 (9.9%)		0.91	0.67–1.25	0.565
Self-employed	526 (15.0%)	413 (14.9%)	113 (15.6)		0.97	0.75–1.26	0.840
Student	146 (4.2%)	93 (3.4%)	53 (7.3%)		0.83	0.56–1.24	0.360
Housewife	94 (2.7%)	88 (3.2%)	6 (0.8%)		1.18	0.47–2.94	0.723
Other	278 (7.9%)	244 (8.8%)	34 (4.7%)		0.69	0.46–1.05	0.083
Unknown	518 (14.8%)	398 (14.3%)	120 (16.5%)		0.99	0.75–1.30	0.929
**HIV RNA, copies/mL, *n* (%)**							
<1000	122 (3.5%)	99 (3.6%)	23 (3.2%)	0.005	0.66	0.39–1.12	0.127
1000–10,000	559 (16.0%)	445 (16.1%)	114 (15.7%)		0.81	0.61–1.08	0.147
10,000–100,000	1470 (42.0%)	1126 (40.6%)	344 (47.4%)		0.94	0.75–1.17	0.562
>100,000	1118 (32.0%)	905 (32.6%)	213 (29.3%)		1.00	-	
Unknown	230 (6.6%)	198 (7.1%)	32 (4.4%)		0.76	0.38–1.55	0.451
**CD4, cells/mm^3^, *n* (%)**							
≤200	791 (22.6%)	721 (26.0%)	70 (9.6%)	<0.001	1.00	-	-
201–500	1485 (42.4%)	1165 (42.0%)	320 (44.1%)		2.22	1.64–2.99	<0.001
>500	1003 (28.7%)	703 (25.4%)	300 (41.3%)		3.01	2.20–4.13	<0.001
Unknown	220 (6.3%)	184 (6.6%)	36 (5.0%)		1.90	0.93–3.90	0.078
**Year of diagnosis,** **median (IQR)**	2011 (2008–2014)	2011 (2007–2014)	2012 (2009–2014)	<0.001	1.09	1.06–1.11	<0.001
**Subtype, *n* (%)**							
A1	104 (3.0%)	85 (3.1%)	19 (2.6%)	<0.001	-	-	-
B	2556 (73.1%)	2038 (73.5%)	518 (71.4%)		-	-	-
C	148 (4.2%)	119 (4.3%)	29 (4.0%)		-	-	-
CRF02_AG	187 (5.3%)	141 (5.1%)	46 (6.3%)		-	-	-
CRF60_BC	64 (1.8%)	12 (0.4%)	52 (7.2%)		-	-	-
F1	179 (5.1%)	157 (5.7%)	22 (3.0%)		-	-	-
Other	261 (7.5%)	221 (8.0%)	40(5.5%)		-	-	-
**Transmitted drug resistance, *n* (%)**							
Any drug class	501 (14.3)	420 (15.1)	81 (11.2)	0.006	-	-	-
NNRTI	335 (9.6)	281 (10.1)	54 (7.4)	0.028	-	-	-
NRTI	150 (4.3)	134 (4.8)	16 (2.2)	0.002	-	-	-
PI	66 (1.9)	53 (1.9)	13 (1.8)	0.832	-	-	-

^a^ By Mann–Whitney test (for quantitative variables) and χ2 test or Fisher’s exact test (for categorical variables), as appropriate. *p*-values <0.05 were considered statistically significant and were reported in bold. ^b^ Adjusted for: Sex, age, mode of HIV transmission, nation of birth, education, employment, plasma HIV-RNA, CD4 cell count, year of diagnosis. Variables that were significant in univariable analysis (*p* < 0.05) were considered for the multivariable model. F: female; IVDU: intravenous drug user; M: male; MSM: men who have sex with men; NNRTI: non-nucleoside reverse transcriptase inhibitor; NRTI: nucleos(t)ide reverse transcriptase inhibitor; PI: protease inhibitor.

**Table 2 viruses-12-00791-t002:** Characteristics of molecular transmission clusters found among 3499 drug-naïve individuals living in Italy.

Cluster’s Characteristics	Small Clusters			Medium Clusters *n* = 36	Large Clusters *n* = 6
	Overall	Clusters of 2 Persons	Clusters of 3 Persons		
	*n* = 186	*n* = 156	*n* = 30		
**Sampling interval (min–max)**	2–3	-	-	4–9	10–52
**Age interval (min–max)**	19–65	19–65	20–56	18–53	18–58
**Mode of HIV transmission, *n* (%)**					
Only MSM	101 (54.3%)	82 (52.5%)	19 (63.3%)	18 (50.0%)	-
Only heterosexual (F+M)	13 (7.0%)	13 (8.3%)	-	-	-
Only heterosexual (only F)	4 (2.2%)	4 (2.6%)	-	-	-
Only heterosexual (only M)	5 (2.7%)	5 (3.2%)	-	-	-
Only IVDU (F+M)	1 (0.5%)	1 (0.6%)	-	-	-
Only IVDU (only M)	2 (1.1%)	2 (1.3%)	-	-	-
Mixed (F+M)	18 (9.7%)	9 (5.8%)	4 (13.3%)	4 (11.1%)	1 (16.7%)
Mixed (only M)	42 (22.5%)	40 (25.5%)	7 (23.3%)	14 (38.9%)	5 (83.3%)
**Nationality, *n* (%)**					
Only Italians	137 73.7%)	122 (78.2%)	15 (50.0%)	24 (66.7%)	2 (33.3%)
Italians and migrants	43 (23.1%)	28 (18.0%)	15 (50.0%)	12 (33.3%)	4 (66.7%)
Only migrants	6 (3.2%)	6 (3.8%)	-	-	-
**Subtype, *n* (%)**					
A1	3 (1.6%)	2 (1.2%)	1 (3.3%)	2 (5.6%)	-
B	149 (80.1%)	125 (80.1%)	24 (80.0%)	25 (69.3%)	3 (50.0%)
C	5 (2.7%)	4 (2.6%)	1 (3.3%)	2 (5.6%)	1 (16.7%)
CRF02_AG	5 (2.7%)	3 (1.9%)	2 (6.7%)	5 (13.9%)	1 (16.7%)
CRF60_BC	-	-	-	-	1 (16.7%)
F1	11 (5.9%)	11 (7.1%)	-	-	-
Other	13 (7.0%)	11 (7.1%)	2 (6.7%)	2 (5.6%)	-
**Resistance, *n* (%)**					
None	150 (80.7)	123 (78.8)	27 (90.0)	31 (86.1)	5 (83.3)
Any drug class	36 (19.3)	33 (21.2)	3 (10.0)	5 (13.9)	1 (16.7)
NNRTI	24 (12.9)	22 (14.1)	2 (6.7)	3 (8.3)	1 (16.7)
NRTI	6 (3.2)	5 (3.2)	1 (3.3)	1 (2.8)	0 (0.0)
PI	7 (3.8)	7 (4.5)	0 (0.0)	1 (2.8)	0 (0.0)

IVDU: intravenous drug user; MSM: men who have sex with men; NNRTI: non-nucleoside reverse transcriptase inhibitor; NRTI: nucleos(t)ide reverse transcriptase inhibitor; PI: protease inhibitor.

**Table 3 viruses-12-00791-t003:** Clusters characteristics.

Medium MTCs (4–9 Sequences)										
ID	Subtype	Cluster Size ^a^ *n*	Sampling Interval Year	HIV Diagnosis Interval Year	Migrants *n* (%)	Migrants’ Nationality *n*	Age, Years Median (IQR)	Risk Factor *n*	Genetic Distance Mean (SE)	Resistance
1	B	4	2006–2007	2006–2007	0 (0.0)	-	46 (44–46)	4 MSM	0.008 (0.002)	No
2	B	4	2007–2012	2004–2012	0 (0.0)	-	46 (42–50)	1 F Het, 2 M Het, 1 M Unk	0.010 (0.002)	No
3	B	4	2008–2014	2008–2014	0 (0.0)	-	52 (50–53)	4 MSM	0.008 (0.002)	No
4	B	4	2008–2016	2008–2016	0 (0.0)	-	39 (37–41)	3 MSM, 1 M Unk	0.006 (0.002)	No
5	B	4	2009–2012	2009–2012	0 (0.0)	-	54 (51–56)	3 MSM, 1 F Het	0.009 (0.002)	No
6	B	4	2009–2016	2009–2016	0 (0.0)	-	36 (33–41)	3 MSM, 1 M Unk	0.009 (0.002)	Yes (*n* = 4; RT: E138A)
7	B	4	2010–2011	2010	1 (25.0)	1 NA	50 (46–52)	4 MSM	0.004 (0.002)	No
8	B	4	2011–2013	2011–2013	0 (0.0)	-	37 (35–38)	4 MSM	0.008 (0.002)	No
9	B	4	2011–2017	2011–2017	0 (0.0)	-	53 (48–56)	4 MSM	0.007 (0.002)	No
10	B	4	2013–2014	2010–2014	0 (0.0)	-	41 (34–48)	4 MSM	0.006 (0.002)	No
11	B	4	2013–2018	2013–2018	1 (25.0)	1 EE	41 (37–43)	3 MSM, 1 M Het	0.008 (0.002)	No
12	B	4	2016	2015–2016	0 (0.0)	-	38 (28–49)	4 MSM	0.004 (0.001)	Yes (*n* = 1; PR: I85V)
13	B	5	2007–2012	2007–2011	1 (20.0)	1 NA	38 (37–45)	5 MSM	0.006 (0.001)	No
14	B	5	2008–2014	2008–2014	0 (0.0)	-	36 (36–37)	5 MSM	0.011 (0.002)	No
15	B	5	2010–2012	2009–2012	2 (40.0)	1 CA, 1 EE	41 (37–43)	5 MSM	0.008 (0.002)	No
16	B	5	2011–2017	2008–2017	0 (0.0)	-	38 (37–45)	4 MSM, 1 M Het	0.009 (0.002)	No
17	B	5	2011–2017	2011–2017	0 (0.0)	-	33 (32–35)	5 MSM	0.009 (0.002)	Yes (*n* = 5; RT: K101E)
18	B	5	2013–2018	2013–2018	1 (20.0)	1 SA	38 (36–43)	3 MSM, 2 M Het	0.011 (0.002)	No
19	B	6	2007–2011	2006–2011	0 (0.0)	-	41 (36–48)	6 MSM	0.010 (0.002)	No
20	B	6	2007–2011	2007–2011	0 (0.0)	-	42 (37–45)	5 MSM, 1 M Unk	0.011 (0.002)	No
21	B	6	2010–2014	2010–2014	0 (0.0)	-	37 (35–43)	5 MSM, 1 M Het	0.011 (0.002)	No
22	B	7	2006–2016	2006–2009	1 (14.3)	1 SA	43 (39–48)	7 MSM	0.010 (0.002)	No
23	B	7	2009–2014	2008–2014	2 (28.6)	2 SA	51 (40–53)	7 MSM	0.011 (0.002)	No
24	B	9	2010–2016	2010–2014	0 (0.0)	-	40 (34–52)	7 MSM, 1 M Het, 1 M Unk	0.009 (0.002)	No
25	B	9	2017–2018	2017–2018	0 (0.0)	-	34 (31–41)	8 MSM, 1 M Unk	0.006 (0.001)	No
26	A1	4	2013–2015	2013–2015	0 (0.0)	-	47 (45–50)	4 MSM	0.010 (0.002)	No
27	A1	8	2014–2016	2013–2016	1 (12.5)	1 WE	37 (33–39)	8 MSM	0.008 (0.002)	No
28	C	4	2007–2011	2006–2011	1 (25.0)	1 SA	46 (44–47)	3 MSM, 1 M Het	0.007 (0.002)	No
29	C	4	2012–2016	2012–2016	0 (0.0)	-	50 (45–52)	3 M Het, 1 M MSM	0.009 (0.002)	No
30	CRF02_AG	4	2009–2013	2009–2012	0 (0.0)	-	35 (35–37)	4 MSM	0.007 (0.002)	No
31	CRF02_AG	4	2009–2013	2009–2013	0 (0.0)	-	38 (38–40)	2 MSM, 1 F Het, 1 M Unk	0.004 (0.001)	No
32	CRF02_AG	4	2014	2014	1 (25.0)	1 EE	34 (30–39)	3 MSM, 1 M Het	0.009 (0.002)	Yes (*n* = 1; RT: E138A)
33	CRF02_AG	5	2013–2014	2012–2014	0 (0.0)	-	34 (33–35)	4 MSM, 1 M Unk	0.006 (0.002)	No
34	CRF02_AG	7	2010–2017	2010–2016	0 (0.0)	-	41 (36–51)	6 MSM, 1 M Het	0.005 (0.001)	No
35	CRF12_BF	5	2014–2015	2013–2015	1 (20.0)	1 SA	34 (27–34)	2 F Het; 2 M Het, 1 M Unk	0.010 (0.002)	Yes (*n* = 3; RT: A62V)
36	CRF20_BG	7	2013–2017	2013–2017	1 (14.3)	1 Aus	33 (32–37)	7 MSM	0.005 (0.001)	No
**Large MTCs (≥10 Sequences)**										
1	B	14	2008–2016	2008–2016	0 (0.0)	-	40 (34–49)	11 MSM, 2 M Het, 1 M IVDU	0.010 (0.001)	No
2	B	19	2007–2015	2007–2015	2 (10.5)	1 EE, 1 SA	37 (32–43)	18 MSM, 1 M Unk	0.016 (0.002)	No
3	B	35	2009–2017	2009–2017	2 (5.7)	1 WE, 1 SA	34 (33–43)	33 MSM, 1 M Het, 1 M Unk	0.013 (0.002)	No
4	C	10	2011–2016	2011–2016	0 (0.0)	-	39 (35–42)	8 MSM, 1 M Het, 1 M Unk	0.009 (0.001)	No
5	CRF02_AG	10	2006–2016	2006–2016	1 (10.0)	1 CA	42 (35–46)	9 MSM, 1 M Het	0.011 (0.002)	No
6	CRF60_BC	52	2008–2018	2008–2018	5 (9.6)	4 EE, 1 Unk	34 (31–37)	41 MSM, 2 F Het, 3 M Het, 6 M Unk	0.012 (0.001)	Yes (*n* = 1; RT: E138A)

^a^ Number of individuals involved in a specific MTC. Het: heterosexual; MSM: men who have sex with men; Unk: unknown; PR: protease; RT: reverse transcriptase. Aus: Australian, CA: Central American; EE: East European; F: female; M: male; NA: North American; SA: South American; WE: West European.

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
