# Peer review of "Evaluation of HIV Transmission Clusters among Natives and Foreigners Living in Italy"

_viruses, 2020, doi:10.3390/v12080791_

Round 1

Reviewer 1 Report

Evaluation of HIV Transmission Clusters Among Natives and Foreigners Living in Italy by Fabeni L et al.

The authors determined HIV-1 molecular transmission clusters (MTCs) among natives and those who were not born in Italy but living in Italy. The study focuses on drug-naïve patients enrolled in the ICONA cohort in the 1998-2018 period.

Comments

Unfortunately, the number of individuals born in Italy is 4 times higher than the number of non-natives, which may strongly affect the MTC clustering.

It would be very important to group the foreigners in different subgroups as it is a very heterogeneous population. Classification according to how long have been living in Italy, their country of origin, immigration status. etc. would give valuable information to monitor the HIV epidemic and to help the most vulnerable populations. Please comment on these issues in the Discussion section.

Author Response

As suggested by the reviewer, we included a table with all the characteristics of migrants, overall and in and out of clusters, including also the information about how long they have been living in Italy. Unfortunately, we don’t have information about the immigration status. We added this new Table to the manuscript as Supplementary Table 1.

Reviewer 2 Report

This manuscript describes characteristics of transmission clusters among HIV-infected people living in Italy, with emphasis on different levels of contribution to clusters by Italian vs. foreign individuals. Factors such as “Italian origin”, “being MSM”, “being young”, and “not being fatal in CD4 counts” significantly contributed to participation in HIV molecular transmission clusters. Furthermore, transmission of drug resistance was less commonly found within clusters than outside of clusters. The results are well presented both in the data and the text. This enables interesting comparison of the given data with previous data on non-B subtypes (Fabeni et al., 2019) or in different countries (Verhofstede et al., 2018). I have some minor concerns as described below.

  1. The Table 1 data look very interesting to me. However, I wonder whether there could be some more information associated with it. For example, I suppose information such as proportions of heterosexual and MSM within each population (Italy, Africa, Central/South America, and Europe) could allow readers to examine the authors’ discussions including the following: “Only six SMTCs of the overall MTCs were composed exclusively of a heterosexual couple of migrant individuals; 67% of them came from the same country in Africa and Asia. This highlights how some migrants probably acquired HIV in their home country and how in certain geographical areas the transmission route is mainly represented by heterosexual contacts” (2nd paragraph of Discussion, page 10).
  2. In the 2nd paragraph of Discussion (page 10), the authors compare their data to the previous data in Belgium, and concludes that they found similar trends in Italy in terms of limited contribution of migrants to HIV transmission clusters. Here, I agree with the authors’ idea, but I suppose it might be interesting to further discuss the differences between the two cohorts.
    • For example, the proportion of foreign people in the Italian cohort was only 19.9 % compared to 53.0 % in the Belgian cohort. As the authors write in the text, this seems to be mainly due to the high prevalence of foreign individuals living in Belgium.
    • Further, foreign HIV+ people (for example those from Europe) might look slightly less actively involved in cluster formation in Italy than in Belgium, although both cohorts showed similar tendencies.
  3. In the 4th paragraph of Discussion (page 11), the authors write “The most important reason for misreporting was fear or embarrassment to disclose sex with same gender sex.” However, I could not find relevant data showing this.
  4. In the 7th paragraph of Discussion (page 11), the authors write “By assessing TDR prevalence in MTCs according to drug class, we found mainly mutations associated with resistance to NNRTIs that are at low fitness cost.” Regarding this, it would be helpful to describe more specifically which mutations are associated with low fitness cost (all?). Adding citations to this sentence could be of further aid.

Author Response

1) The Table 1 data look very interesting to me. However, I wonder whether there could be some more information associated with it. For example, I suppose information such as proportions of heterosexual and MSM within each population (Italy, Africa, Central/South America, and Europe) could allow readers to examine the authors’ discussions including the following: “Only six SMTCs of the overall MTCs were composed exclusively of a heterosexual couple of migrant individuals; 67% of them came from the same country in Africa and Asia. This highlights how some migrants probably acquired HIV in their home country and how in certain geographical areas the transmission route is mainly represented by heterosexual contacts” (2nd paragraph of Discussion, page 10).

Answer: Detailed information of some characteristics such sex, country of origin and risk factor for medium and large clusters are reported in Table 3, which are in our opinion the most interesting clusters. Concerning the small clusters, we tried to provide some more information in Table 2 and in Figure 1, where risk factor and country of origin of all individuals considered for this study are reported. In this Figure in particular, the 6 couples mentioned in the discussion are now clearly evident. In this regard, we would like to thank the reviewer, because, in answering his/her question, we realized that the sentence was not complete and incorrect. In fact, the correct sentence is the following “Only six (3.2%) of the SMTCs were totally composed of migrant couples (Table 2) originating from Africa (n=3), Asia (n=1), America (n=1) and Eastern Europe (n=1) (data not shown); four of them (67%) were composed of heterosexual individuals that came from the same country in Africa (n=3) and Asia (n=1) (Figure 1), while the other two were composed of MSM.” We have now reported this sentence in the Results Section, paragraph “3.5 Characteristics of the small clusters” at page 8. Whereas, in the results section we reported only a summary of this sentence.”.”

2) In the 2nd paragraph of Discussion (page 10), the authors compare their data to the previous data in Belgium, and concludes that they found similar trends in Italy in terms of limited contribution of migrants to HIV transmission clusters. Here, I agree with the authors’ idea, but I suppose it might be interesting to further discuss the differences between the two cohorts.

- For example, the proportion of foreign people in the Italian cohort was only 19.9 % compared to 53.0 % in the Belgian cohort. As the authors write in the text, this seems to be mainly due to the high prevalence of foreign individuals living in Belgium.

- Further, foreign HIV+ people (for example those from Europe) might look slightly less actively involved in cluster formation in Italy than in Belgium, although both cohorts showed similar tendencies.

Answer: As suggested by the reviewer, the discussion now also reports the differences between the two cohorts. Therefore, the second paragraph is now structured as follows: “A minimal contribution of migrants in MTCs was also found in Belgium, despite the high prevalence of foreign individuals living in this country [10]. In particular, in the Belgian cohort the percentage of foreign people was much higher than in our cohort (53.0% vs. 19.9%). Furthermore, foreign individuals infected with HIV (for example those from Europe) might look slightly less actively involved in cluster formation in Italy than in Belgium, although both cohorts showed similar tendencies. Despite these differences between the two cohorts, overall findings support the fact that clustering is mainly driven by natives.

3) In the 4th paragraph of Discussion (page 11), the authors write “The most important reason for misreporting was fear or embarrassment to disclose sex with same gender sex.” However, I could not find relevant data showing this.

Answer: The sentence is related to experience reported by clinician’s in their clinical routine. However, as no data are available regarding this point, following the reviewer’s suggestion, we softened the sentence as follows: “The most important reason for misreporting could be due to fear or embarrassment to disclose sex with same gender sex.”

4) In the 7th paragraph of Discussion (page 11), the authors write “By assessing TDR prevalence in MTCs according to drug class, we found mainly mutations associated with resistance to NNRTIs that are at low fitness cost.” Regarding this, it would be helpful to describe more specifically which mutations are associated with low fitness cost (all?). Adding citations to this sentence could be of further aid.

Answer: The two NRRTI mutations found in MTCs are K101E and E138K. Therefore, as suggested by the reviewer, we now mention these mutations in the sentence as follows “…we mainly found the mutations associated with resistance to NNRTIs K101E and E138A, which are at low fitness cost”. Moreover, we reported the following references about their viral fitness:

Rhee, S. Y., Blanco, J. L., Jordan, M. R., et al. (2015). Correction: Geographic and Temporal Trends in the Molecular Epidemiology and Genetic Mechanisms of Transmitted HIV-1 Drug Resistance: An Individual-Patient- and Sequence-Level Meta-Analysis. PLoS medicine, 12(6), e1001845.

Machnowska, P., Meixenberger, K., Schmidt, D., et al., & German HIV-1 Seroconverter Study Group (2019). Prevalence and persistence of transmitted drug resistance mutations in the German HIV-1 Seroconverter Study Cohort. PloS one, 14(1), e0209605.

Reviewer 3 Report

This study used pol protease-RT region sequences from HIV-infected people sampled in Italy, to assess transmission clusters.

1) I do not like the term "molecular transmission clusters", but if this term is the common or accepted term for the inference of transmissions based on DNA sequences, it is acceptable to me. 

2) The sequences must be submitted to GenBank, and the accession numbers listed in the text of the paper.  The GenBank entries need to contain annotation for date of sample, country of sample, and subtype of virus at a minimum.  You have risk factor, viral load and other data which should also be annotated, but if your IRB does not allow that data to be included, it can be omitted.

3) A histogram or some other representation of the pairwise distances should be included in the figures.  Although the 1% distance as a cutoff for defining inside or outside a "cluster" is indeed the standard for HIV-TRACE and some similar programs, this is a moving target over time as HIV continues to diversify, and also the optimal standard in one country or even city, may be different than the optimal standard in another, depending on how long the epidemic has been ongoing in the region and on what proportion of IVDU or nosocomial outbreaks have influenced the local epidemic.

The PASC tool at NCBI, or a similar tool, can be used to produce a histogram of pairwise distances.  But it may take some work to use it for your own data.  https://www.ncbi.nlm.nih.gov/pmc/articles/PMC4221606/

Alternatively, a phylogenetic tree could be used to illustrate the diversity in your data, or at least make a tree that can be viewed in a supplement.

4) It would be very interesting to not geographic distances within the clusters within Italy.  Most human subjects review boards do not want city level identification of each sequence listed in GenBank, for confidentiality reasons, but noting whether the clusters were mostly form the same city, or alternatively if the clusters tend to indicate that people travel widely within Italy, would be very important for prevention efforts. 

5) Figure 1, the HIV-TRACE diagram, indicates two very large clusters and a few more large clusters.  The text discusses these only very briefly.  It would be very interesting for example to know if these are clusters that started in the 1990's and are large because they started early and spread slowly, or instead if they represent some local rapid spread or "superspreader" event(s).  Likewise it would be interesting to know the subtype of virus involved. 

Not all of this (subtype, city, date, etc) can be plotted in one figure of course, but the authors could explore this a bit and at least note in the text if there were or were not any interesting findings.  For example they might find and state that "the 3 largest clusters were from 2 major cities and one of them was from a slow prolonged spread while the other seems to represent a recent explosion of cases". or something like that.

Author Response

1) I do not like the term "molecular transmission clusters", but if this term is the common or accepted term for the inference of transmissions based on DNA sequences, it is acceptable to me.

Answer: We used the term “molecular transmission clusters” in accordance to CDC and other works focused on the same topic (please see some references below). In particular, CDC uses “molecular clusters” or “molecular transmission clusters” to indicate clusters of HIV infections with closely related strains. CDC reported that a molecular cluster is a group of persons with diagnosed HIV infection who have genetically similar HIV strains (https://www.cdc.gov/hiv/pdf/funding/announcements/ps18-1802/CDC-HIV-PS18-1802-AttachmentE-Detecting-Investigating-and-Responding-to-HIV-Transmission-Clusters.pdf).

The following papers offer examples of works that used the term molecular transmission clusters or networks:

  • Paraskevis D, et al. HIV-1 molecular transmission clusters in nine European countries and Canada: association with demographic and clinical factors. BMC Med. 2019;17(1):4;
  • Wertheim JO, et al. Incident infection in high-priority HIV molecular transmission clusters in the United States. AIDS. 2020;34(8):1187-1193;
  • Chan PA, et al. Phylogenetic Investigation of a Statewide HIV-1 Epidemic Reveals Ongoing and Active Transmission Networks Among Men Who Have Sex With Men. J Acquir Immune Defic Syndr. 2015;70(4):428-435;
  • Kosakovsky et al. HIV-TRACE (transmission cluster engine): a tool for large scale molecular epidemiology of HIV-1 and other rapidly evolving pathogens. Mol Biol Evol 2018;35:1812–9;
  • Beloukas A, et al. Molecular epidemiology of HIV-1 infection in Europe: An overview. Infect Genet Evol. 2016;46:180-189.

2) The sequences must be submitted to GenBank, and the accession numbers listed in the text of the paper.  The GenBank entries need to contain annotation for date of sample, country of sample, and subtype of virus at a minimum.  You have risk factor, viral load and other data which should also be annotated, but if your IRB does not allow that data to be included, it can be omitted.

Answer: All 726 sequences involved in the 228 molecular transmission clusters were submitted into GenBank on Friday 26 June (ID submission: 2358678). We received an email from ncbi that reported that we should receive GenBank accession numbers for our sequences generally within two working days, unless there are issues with our submission that must be resolved first. We cannot provide this information for the revision due to time limitation for the revision (five days). However, we are confident that at the moment of the final publication the accession number will be available. We selected the variables to be annotated according to the most common Source Modifiers, reported on the site https://www.ncbi.nlm.nih.gov/WebSub/html/help/genbank-source-table.html#modifiers. In particular, the following variables have been reported: Type of isolate; isolation source; host; subtype; collection_date; country; sex, protein, gene. We reported the following sentence regarding the submitted sequences in the text: “All 726 sequences involved in the 228 molecular transmission clusters have been deposited into GenBank. All other sequences are available on request.” (see paragraph “2.6 Sequences deposited to GenBank”, page 5).

3) A histogram or some other representation of the pairwise distances should be included in the figures.  Although the 1% distance as a cutoff for defining inside or outside a "cluster" is indeed the standard for HIV-TRACE and some similar programs, this is a moving target over time as HIV continues to diversify, and also the optimal standard in one country or even city, may be different than the optimal standard in another, depending on how long the epidemic has been ongoing in the region and on what proportion of IVDU or nosocomial outbreaks have influenced the local epidemic.

The PASC tool at NCBI, or a similar tool, can be used to produce a histogram of pairwise distances.  But it may take some work to use it for your own data.  https://www.ncbi.nlm.nih.gov/pmc/articles/PMC4221606/

Alternatively, a phylogenetic tree could be used to illustrate the diversity in your data, or at least make a tree that can be viewed in a supplement.

Answer: As suggested by the reviewer, phylogenetic trees of both the entire dataset of 3499 pol sequences coloured according to subtypes and of the 726 pol sequences involved in 228 molecular transmission clusters are now present as supplementary figures 1 & 2.

4) It would be very interesting to note geographic distances within the clusters within Italy.  Most human subjects review boards do not want city level identification of each sequence listed in GenBank, for confidentiality reasons, but noting whether the clusters were mostly from the same city, or alternatively if the clusters tend to indicate that people travel widely within Italy, would be very important for prevention efforts.

Answer: We completely agree with the reviewer. However, to correctly answer his/her question we would not only need the city where the sequence test was performed but also the city of residence and not all the patients are diagnosed and followed in the same city they live in. Unfortunately, we don’t have the city of residence, as this represents a sensible datum. On the other hand, our goal was to have a general overview of the transmission clusters found in newly diagnosed individuals infected with HIV-1 living in Italy, and to understand the contribution of migrants in these transmission clusters.

5) Figure 1, the HIV-TRACE diagram, indicates two very large clusters and a few more large clusters.  The text discusses these only very briefly.  It would be very interesting for example to know if these are clusters that started in the 1990's and are large because they started early and spread slowly, or instead if they represent some local rapid spread or "superspreader" event(s).  Likewise it would be interesting to know the subtype of virus involved. 

Not all of this (subtype, city, date, etc) can be plotted in one figure of course, but the authors could explore this a bit and at least note in the text if there were or were not any interesting findings.  For example they might find and state that "the 3 largest clusters were from 2 major cities and one of them was from a slow prolonged spread while the other seems to represent a recent explosion of cases". or something like that.

Answer: Regarding the question of the origin of the clusters, we will evaluate this parameter in future studies. Even though we are conscious that the information about timing could enrich the manuscript, the analysis required is laborious and time consuming, and due to time limitation for the revision (five days), the requested analysis cannot be included at this time.